# Polymeric/Dextran Wafer Dressings as Promising Long-Acting Delivery Systems for Curcumin Topical Delivery and Enhancing Wound Healing in Male Wistar Albino Rats

**DOI:** 10.3390/ph16010038

**Published:** 2022-12-27

**Authors:** Adel Al Fatease, Mohammed A. S. Abourehab, Ali M. Alqahtani, Kumarappan Chidambaram, Absar Ahmed Qureshi, Krishnaraju Venkatesan, Sultan M. Alshahrani, Hamdy Abdelkader

**Affiliations:** 1Department of Pharmaceutics, College of Pharmacy, King Khalid University, Abha 62529, Saudi Arabia; 2Department of Pharmaceutics, Faculty of Pharmacy, Minia University, Minia 61519, Egypt; 3Department of Pharmaceutics, College of Pharmacy, Umm Al-Qura University, Makkah 21955, Saudi Arabia; 4Department of Pharmacology, College of Pharmacy, King Khalid University, Abha 62529, Saudi Arabia; 5Department of Clinical Pharmacy, College of Pharmacy, King Khalid University, Abha 62529, Saudi Arabia

**Keywords:** curcumin, dextran, wafer, wound, alginate, dressing

## Abstract

Curcumin is the main active constituent in turmeric, and it is one of the biopolyphenolic compounds. A cumulative body of research supports the use of curcumin in the treatment of wounds, yet poor water solubility and lack of therapeutic dose determination hamper its use for this therapeutic purpose. This work aimed at preparing novel curcumin wafer dressings to provide a favorable environment for wound healing. Hybrid synthetic (PVA, PVP, HPMC, and CMC) and biodegradable (sodium alginate and dextran) polymers were employed to prepare wafer dressings loaded with incremental three doses (2, 10, and 20 mg) of curcumin per a wafer dressing. The solvent casting method was used to prepare the dressings. Dimension, surface pH, mechanical properties, DSC, FTIR, XRD, erosion time, and in vitro release were studied. Skin wound healing assay was studied in Wistar albino rats. Six curcumin-loaded wafers were successfully prepared with good mechanical properties. Curcumin was dispersed in an amorphous/molecular form, as evidenced by thermal (DSC) and spectral (FTIR and XRD) analyses. Prolonged curcumin release (>24 h) was recorded for F6 (10 mg curcumin) and F7 (20 mg curcumin). Wound healing rate constants and time for 50% wound closure (T1/2) were estimated from a semi-log wound diameter versus time curve. A superior healing rate (up to 3-fold faster) was recorded for curcumin-loaded wafer dressings containing 10 mg (F6) with T1/2 of 7 days compared to 20 days for the placebo-treated group. These results warrant using the selected curcumin-loaded wafer dressing for safer and faster wound closure.

## 1. Introduction

The wound is an injury or disruption of the skin’s normal structure, barrier, and physiological function [1]. Impaired and delayed skin wound healing might result in infection, tissue necrosis, gangrene, and periwound dermatitis [2].

A vascularized wound area is essential for a healthy wound and healing process. Excision of any dead tissue, sterile (no microbes), and moist wound are required for normal wound healing. Wound dressings have been shown to enhance the wound healing process by absorbing exudate, inhibiting bacterial growth, ensuring fluid balance, and being easy to use by the patient or health care provider [3]. Innovation in topical skin and transdermal drug delivery has been advancing to improve drug permeability, compared to poor permeability from conventional ointments and creams. Transdermal patches have been around for delivering many drugs (e.g., fentanyl, nitorglycerin, estradiol, and nicotine) to systemic circulation. On the contrary, new advancements in topical skin delivery described polymeric wafers that are composed of semisynthetic cellulose polymer (e.g., carboxymethyl cellulose), gelatin, and alginate sodium to provide a moist and favorable occlusive environment for wound healing [4]. Typical hydrocolloid dressings are composed of outer backing layers for fixing on-site and therapeutic inner gelling wound coating layers [5].

Hydrophilic polymer dressings or hydrocolloid dressings can be prepared in the form of wafers, powders, or pastes. These dressings might be composed of hydrophilic colloids such as gelatin, pectin, and carboxymethyl cellulose. Polymeric dressings can offer a moist environment for wound healing and a physical barrier against invading bacteria. Wafer dressings can offer sustained protection and be kept in place for a longer time than creams or ointments containing silver diazine [4].

This unique feature for long-acting polymeric dressings can offer many advantages for patients and clinicians. Polymeric wound dressings can absorb excessive exudates; they can reduce the need for the daily care of and cleaning of the wound from exudates and reduce the number of visits to the surgeons. Polymeric wafer dressings can lend physical barriers to provide protection of the wound against microbial invasion, leading to an enhanced wound healing process. Polymeric wafer dressings can be loaded with some wound-promoting agents and antibiotics [6,7,8].

Polymeric wafer dressings are essentially made of two parts: a hydrocolloid polymeric layer and a backing (outer layer). When placed on the wound, the ideal wound dressing provides warmth that can soften and transform into a gel coverage for the wound bed. Hydrophilic polymer-based dressings can absorb appreciable amounts of exudates; these dressings can be suitable for rehydrating dry necrotic tissues. Hydrocolloid dressings are available in many shapes, and some also have an additional adhesive border to prevent the dressing from sliding over the wound [5]. Polymeric wafer dressings loaded with drugs can exhibit high porosity, act as scaffolds, promote cellular granulation and proliferation and allow gas exchange during the wound healing process [9,10].

Biosurfactant-loaded wafer dressings made of two polysaccharides, sodium alginate and carrageenan, have been successfully prepared and characterized for morphology, wound adhesion, and mechanical properties [6].

However, current wound dressings suffer from shortcomings such as poor mechanical, antimicrobial properties, and breaking during patient handling [10]. While maintaining the typical features of dressings/wafers as thin and delicate, these formulations must also be strong enough to resist breakage when handled and applied by a patient or a member of the medical staff.

The addition of some synthetic polymers, such as polyvinyl alcohol and polylactic glycolic acid, to natural polysaccharides has been reported to improve drug loading capacity and scaffolding properties [9,10]. Therefore, combining synthetic polymers and natural biopolymers such as dextran and sodium alginate can offer wound dressing with a good mechanical and absorbing capacity [8]. Sodium alginate is a typical example of a biopolymer that has been used in wound dressings for biocompatibility and non-toxicity [11].

Curcumin is a biopolyphenolic compound, and it is the main active constituent in turmeric. A cumulative body of research supports the use of curcumin in the treatment of wounds; this is because curcumin has anti-inflammatory, antiglycation, antioxidant and antibacterial properties that can promote wound healing [8,12]. Several studies have been reported on the enhancing wound healing properties of curcumin; however, scarce reports are hardly found for determining the exact dose or loading capacity of curcumin per patch or wafer dressings [12,13,14,15]. The solubilizing capacity of the vehicle (polymeric wafer dressings) for the poorly soluble curcumin will also be investigated.

In this study, a combination of synthetic polymers, such as polyvinyl pyrrolidone and polyvinyl alcohol, and natural biopolymers, such as dextran, sodium alginate, hydroxypropyl methyl cellulose, and carboxymethyl cellulose, were employed. The loading capacity, mechanical properties, surface morphology, pH, in vitro drug release of the polymeric wafer dressings, and in vivo evaluation using an appropriate animal wound model for wound healing will be studied.

## 2. Results and Discussion

### 2.1. Preparation of Curcumin-Loaded Wafer Dressings

Our design was based on the preparation of seven different wafer dressings prepared from different combinations of synthetic polymers (PVA and PVP); and natural hydrophilic polymers (SA, HPMC, and CMC) using the solvent casting method. This method has been successful in the preparation of many wound dressings because of its simplicity and robustness [16]. The wafer dressing-forming PVA is known for its film-forming properties, and PVP was used as a solubilizing agent for the insoluble drug curcumin [17]. Other hydrocolloids such as SA, HPMC, and CMC are mucoadhesive polymers that can allow the dressings to adhere firmly to the wound area, providing sustained release characteristics and localizing the curcumin effects. Other excipients, such as PEG 1000 and glycerine, were employed as plasticizers [18]. Dextran is a branched high molecular weight polymer of glucose units [19]. Dextran is commonly used in wound dressings for mucoadhesion, bulking agents, and high-water binding capacity [19,20]. Four different wound dressings (F1–F4) composed of the same concentrations of the film-forming polymer (PVA) and four different mucoadhesive polymers (HPMC, CMC, SA, and PVP) were prepared. A combination of the two synthetic polymers, PVA and PVP (F4), cast brittle wafers with very poor mechanical properties that did not allow cutting the wafers with uniform sizes; therefore, they were excluded from further characterization. Drug content was found to be in the range of 1.85 to 2 mg for F1 to F5; the weight of curcumin wafer dressings ranged from 112 to 118 mg, respectively. This indicated the uniformity and reproducibility of the method adopted. Similarly, the recorded sickness of the prepared wafer dressings was in the range of 200 to 2010 µm with less than a 5% relative standard deviation.

F5 was successfully prepared from the combination of the two synthetics and the three biopolymers. Better mechanical strengths and sufficient flexibility were achieved for F5. In order to assess the loading capacity of this optimized wafer dressing, 5-fold and 10-fold increments of curcumin from the same combination were reproduced to form F6 and F7, respectively. Wafer dressings (F6 and F7) with greater doses of curcumin 5 mg- and 10 mg-wafer were fabricated and exhibited good loading capacity to accommodate up to 10-fold greater doses of the insoluble drug curcumin to allow therapeutic dose finding.

### 2.2. Characterization of Curcumin Loaded Wafer Dressings

Six out of seven curcumin-loaded wafer dressings were successfully prepared using the solvent casting method. Figure 1A shows some curcumin loaded wafer dressings; Table 1 shows the dimensions: weight (mg) and thickness (µm) of the prepared formulations. The weight of the prepared wafer dressings ranged from 112 to 135 mg; the thickness was 200 to 325 µm. Curcumin content measured for the prepared wafer dressings was in the range of approximately 2 to 20 mg. F5, F6, and F7 contained 2, 10, and 20 mg, respectively, representing wafer dressings that had the same composition, but they contained different loading capacities of 1-fold, 5-fold, and 10-fold increments, respectively. The pH values of the prepared curcumin-loaded dressings recorded pH 7 using the pH indicator flexible strips and validated using two standards: acid (stained red color) and alkali (stained blue color) Figure 1B. These results indicate that the prepared curcumin wafer dressings could pose no harmful or irritant effects when applied on disruptive skin tissues.

Folding endurance is a measure of mechanical properties and was correlated well with elasticity and tensile strength [21]. The folding endurance measured for the prepared dressing was 3 to 10. The lowest number recorded for F3 indicated poor elasticity and mechanical properties for alginate-based wafer dressing [11], while other wafer dressings based on a combination of the natural, semisynthetic and synthetic polymers showed reasonable mechanical properties that could resist handling and clinical relevance. HPMC demonstrated good mechanical properties; however, it has been previously reported to show relatively poor mucoadhesion characteristics, compared to SA and CMC due to partial methylation of free hydroxyl groups and absence of carboxylate ionizable groups involved in hydrogen bonding and electrostatic attraction with mucin [17,21]. Different blends using a combination of SA, CMC, and PVP generated wafer dressings with better mechanical properties and greater loading capacity to accommodate the insoluble drug up to 10-fold drug concentrations.

Similarly, the bursting force (toughness) and extensibility (elongation), as shown in Figure 1C, were recorded for the prepared wafer dressings; while there were significant differences (*p* < 0.01) recorded for toughness measured to the prepared wafer dressings, no statistically significant differences (*p* > 0.05) recorded for extensibility. These results could indicate that F1 and F2 (HPMC and CMC-based wound dressing, respectively) were tougher and more cohesive compared to other combination-based wafer dressings. When it came to elasticity and tensile strength, no marked differences were recorded for all prepared dressing formulations. The erosion time measured for the prepared curcumin wafer dressings ranged from 5 to 30 min; rapid erosion time was measured for F3 and F5, while longer erosion time of up to 30 min was recorded for F1, F2, F6, and F7. These results indicated that erosion time was significantly (*p* < 0.05) dependent on the polymeric composition and drug content of the insoluble drug curcumin. A higher concentration of hydrophobic insoluble drug was likely to increase hydrophobicity and decrease the wettability of the prepared wafers; SA and a combination of SA and other polymers enhanced wettability and erosion time. In addition, erosion time correlated well with the mechanical properties; the lower the folding endurance, the shorter the erosion time was recorded for the wafer dressings. Our hypothesis is that rapid wettability and erosion could result in faster drug release, better wetting of the wound, and prevention of excessive dryness of the wound. These could benefit the patient and wound healing process [3]. Therefore, F1, F5, F6 and F7 were selected for further investigation

### 2.3. XRD, DSC and FTIR Studies

Figure 2 shows the XRD diffractograms of curcumin, a drug-free wound dressing, and four selected curcumin-loaded wafer dressings representing two curcumin wafers (F1 and F5) of different compositions and three wafers (F5, F6, and F7) containing three different doses of curcumin (1-fold, 5-folds, and 10-folds of curcumin loading). Curcumin diffractogram showed strong diffraction peaks, indicating a highly crystalline solid powder. Drug-free wafers showed a typical amorphous with very weak signals and halo patterns, indicating the presence of an amorphous polymeric system [22]. More interestingly, curcumin-loaded wafers did not show the characteristic X-ray diffraction pattern of curcumin powder. This result indicated that the drug converted into an amorphous form in the prepared wafer dressings irrespective of the composition of the wafer dressing or the drug loading concentrations up to 10-fold increments.

DSC thermograms of curcumin powder and the selected curcumin-free and loaded wafers, as abovementioned, are shown in Figure 3. A strong endothermic peak was recorded at 182 °C for curcumin; this indicates the melting of curcumin [23,24]. The drug-free wafer F5 showed no thermal event indicating amorphousness of the polymeric wafer. The broad shallow endothermic peak appeared typically from 60 °C to 120 °C) was due to loss of moisture (bound water) from the chains of the hydrophilic polymers [25]. The DSC thermograms collected for the selected curcumin-loaded wound dressing wafers showed the complete disappearance of the endothermic melting peak. These results indicated that curcumin was dispersed in an amorphous form/molecular dispersion within the prepared wafers and resulted in solid solutions. These results are correlated well with XRD patterns described above.

Figure 4 shows FTIR spectra of curcumin powder, drug-free wafers, and the selected curcumin-loaded wafers F1 and F5-F7. The FTIR spectra of curcumin free wafer composed of PVA, alginate, CMC, HPMC, and dextran showed a broad peak at 3500 cm^−1^ indicating stretching of the hydroxyl (-OH) groups, and the stretching vibration of methyl, carbonyl and -CH from 1600–1700 cm^−1^. The FTIR spectrum of curcumin demonstrated bands of the phenolic OH group that stretches at 3510 cm^−1^ [26]. Other bands at 1620 cm^−1^ were due to the stretching vibration of carbonyl and aromatic [27]. These characteristic FTIR bands of curcumin disappeared in the collected spectra for F1 and F5–F7 indicating molecular electrostatic attraction, hydrogen bonding, and wander Waals forces formation between the drug and the polymeric wafer dressings. Similar results have been reported elsewhere; diclofenac and streptomycin were loaded into hydrophilic wound dressings in a molecular dispersion. The hydrophilic wound dressings were prepared using the solvent casting method and were composed of a blend of HPMC, SA, and carrageenan and plasticized with glycerol [16]. The molecular dispersion of drugs in the wound dressings was confirmed by DSC, FTIR, and XRD studies.

### 2.4. In Vitro Release of Selected Curcumin-Loaded Wafer Dressings

Four (F1, F5, F6, and F7) curcumin-loaded wafer dressing formulations were selected for in vitro release studies (Figure 5). The selected formulations were composed of different polymer blends (F1 and F5) and increasing amounts of curcumin (x-, 5x- and 10x-folds for F5, F6, and F7, respectively), as shown in Figure 5. All wafer dressings demonstrated a lag time (no release) of up to 1 h, followed by slow and steady curcumin release over 8 h. Only F1 and F5 recorded complete drug release, and it was obtained over a period of 24 h. On the contrary, F6 and F7 showed significantly more prolonged drug release, up to 30% over a period of 24 h. The composition of wafer dressings (F1 and F5) did not generate statistically significant (*p* > 0.05) differences in curcumin release. Higher drug loading (F7) resulted in up to 3-fold decreases in curcumin release rates, compared to F5 with the same composition (Table 1). These results could indicate that drug loading had a significant effect (*p* < 0.01) on drug release rates. This could be explained on the basis that the higher drug load of the hydrophobic drug curcumin reduces wettability, erosion time, and hence drug release. Consistent and prolonged curcumin release over a period of 24 h from the prepared wound wafer dressings could be advantageous. This is because the wafer dressing would provide the wound with the healing agent over an extended period for better wound healing [16,28].

It is worth mentioning that Tween 80 was added to the release medium containing phosphate buffer pH 7.4 to ensure the wettability of the prepared formulation loaded with the poorly soluble drug and to maintain sink conditions throughout the release experiment. Drug release from curcumin suspension was relatively slow and did not achieve a percentage greater than 3% over the release period.

The cumulative release data were fitted to four kinetics models (zero order, first order, Higuchi, and Korsmeyer–Peppas models). The release rate constants and regression coefficients were recorded. As shown in Table 2, the best-fitting kinetics model was the Higuchi model, as the regression coefficient indicated. This means that the general release mechanism of the poorly soluble drug from the prepared wafer dressings was diffusion from the polymeric matrix. The release rate constants (K_H_) correlated well with the in vitro release profiles. The lowest release rate constant was recorded for F7, as mentioned above.

### 2.5. Skin Wound Healing Studies

The wound sizes and reduction of wound sizes were assessed by measuring the wound diameter for placebo (drug-free wafer dressing), curcumin suspension, and three different curcumin-loaded wafer dressings (F5, F6, and F7) over a period of 12 days, as shown in Figure 6 and Figure 7, respectively. Similar enhanced wound healing time (14 days) has been reported with polyurethane/siloxane dressings loaded with renewable raw materials having antioxidant and antimicrobial properties, compared to delayed wound healing time (>20 days) for the untreated control group [29]. As previously outlined, both antioxidant and antimicrobial properties have been reported for curcumin [8].

The three different formulations (F5–F7) had the same polymeric composition but three different increments of curcumin loading (2 mg, 10 mg, and 20 mg) in order to find the most suitable dose of curcumin for fast wound healing. Table 3 shows the wound healing rate constants, the time for a 50% decrease in wound size (T_1/2_), and linearity/regression coefficients from first-order kinetic semi-logarithmic curves of log diameter versus time (days).

The fastest wound closure rate was recorded for F6, which contained 10 mg of curcumin per wafer dressing. Furthermore, the T_1/2_ recorded was the fastest (6 days) among placebo- and curcumin-loaded wafer dressings with lower (2 mg) and higher (20 mg) doses. Curcumin suspension and curcumin-loaded wafer dressings showed statistically significant (*p* < 0.05) differences in both healing rate constants and T_1/2_ values from the placebo-treated group.

The mathematical model was employed to measure the wound healing rates with the main objective of quantifying the effect of the wound healing agent [30,31]. The healing rate constant for F6 (10 mg curcumin) showed a faster healing rate by 1.6 times than that containing a lower concentration from F5 (2 mg). On the contrary, further increases in curcumin amounts (20 mg) slightly but non-significantly (*p* > 0.05) decreased the healing rate and healing time. The optimum concentration for enhanced wound healing appeared to be 10 mg/wafer dressings. Curcumin has been reported to reduce inflammation by inhibiting the nuclear factor (NF-kB) and suppressing TNF expression. Curcumin can exert its anti-inflammatory effects by acting on other signaling pathways, such as peroxisome proliferator-activated receptor-gamma, and by lowering reactive oxygen species (ROS) levels [14]. In addition, curcumin has been reported to have antibacterial properties that can inhibit bacterial growth in the wound area [12]. These results could confirm the notion that occlusive dressings increase cell proliferation and activity by keeping an optimum level of exudates, protecting the wound from further trauma, and avoiding excessive tissue maceration, compared to ointments and creams [28,32]. Under occlusive conditions, hydrophilic colloids-based wound dressings form a gel on the surface of the wound and provide moist wound healing [28].

## 3. Materials and Methods

Curcumin 97% was supplied from Alfa Aesar, Thermo Fisher Scientific, Heysham, Lancashire, UK. Polyvinylpyrrolidone K15 (PVP), polyethylene glycol (PEG) 1000, glycerine, and carboxymethyl cellulose (CMC) (viscosity 1% equal to 400–1000 cps) were purchased from Fluka, GmbH, Buchs, USA. Sodium alginate (SA) low viscosity grade was purchased from Acros Organics, Morris, NJ, USA. Hydroxypropyl methylcellulose (HPMC) E 5 LV was purchased from Loba Chemie, Mumbai, India. Polyvinyl alcohol (PVA) molecular weight (MW) 14,000 was supplied by BDH Chemicals Ltd., Poole, England. Dextran (High Fraction) was purchased from Eastman Kodak Company, Rochester, NY, USA. Cellulose membrane MW 12,000–14,000 Da was purchased from Sigma-Aldrich, USA. Silk pool plaster (backing adhesive plaster) Sensitive Fix^®^, Pic solution, Artsana Group, Italy.

### 3.1. Preparation of Curcumin-Loaded Wafer Dressings

Accurate amounts of PVA, PVP, SA, CMC, and HPMC polymers were weighed and dissolved in distilled water on a hot plate stirrer (Hotplate Stirrer, LabTech, Daihan, Korea) adjusted at 80 °C for 2 h to make up 5%, 10, 5%, 5% and 5% *w*/*w* polymer solutions. The prepared polymer solutions were allowed to cool and set by storage overnight in a fridge at 4–8 °C.

Accurate amounts of curcumin, plasticizers (PEG 1000 and glycerin), dextran, and aliquots of the prepared polymer solutions were weighed in 100 mL beakers according to the composition shown in Table 4. The mixtures were sonicated in a bath sonicator (Elmasonic E120 H, Elma, Bedford, UK) for 5 min and then stirred on the hot plate (Hot-plate Stirrer, LabTech, Daihan, Korea) for another 5 min. The mixtures were transferred into a 9 cm diameter Petri dish and left to dry for 48 h. The wafer dressings were prepared under aseptic conditions, and the polymeric mixtures were allowed to dry in pre-sterilized cabinets. The casted polymeric dressings were cut into circular discs by using a cork borer (20 mm in diameter). A graphical summary of the solvent casting method is outlined in Figure 8. F5 was loaded with increasing amounts curcumin up to 5- and 10-fold increments for F6 and F7, respectively, and prepared as mentioned above.

### 3.2. Characterization of Curcumin Wafer Dressings

#### 3.2.1. Dimension, Drug Content, pH Measurements

Weight, thickness, drug content, folding endurance, and in vitro erosion time were studied as previously reported [14]. The thickness was measured using a Metric External micrometer, Draper, Hampshire, UK. The weight of individual wafer dressings was measured using an analytical balance (Ohaus^®^ Adventurer^®^ Analytical Balance, Mettler Toledo, Zurich, Switzerland). For drug content, six wafer dressings were dispersed in 20 mL of sodium hydroxide (1 M), sonicated for 15 min, filtered and appropriately diluted with sodium in the hydroxide solution water, and measured spectrophotometrically at 470 nm using Shimadzu UV-2550 Spectrophotometer, Kyoto, Japan. The surface pH of the prepared curcumin dressings was measured using pH indicator papers, Universal pH (0–14), Fisher Scientific, London, UK. A sample of each curcumin wafer dressing was placed on a small porcelain dish. An aliquot of 1 mL of distilled water was added to the dressing and was allowed to hydrate for 5 min; pH indicator flexible strips were employed to be immersed in the formed gel and to measure pH by matching the developed color with the standard color bands from pH 0 to 14 to interpret the pH of the polymeric wafer dressings.

#### 3.2.2. Folding Endurance and Mechanical Properties

Curcumin wafer dressings were frequently folded from the central till the polymeric wafer dressings were torn. The number of times the curcumin wafer dressings bent until tearing was the value of folding endurance. Toughness (mN) and extensibility (mm) are the force of wafer dressings, and ability to stretch/extend on break is measured using TA. XT plus texture analyzer, Stable Micro Systems, Surrey, UK. The measurements were performed using a measure force in compression mode and 5 mm spherical probe. The measurement parameters were pre-test speed (1 mm/s), test speed (1 mm/s), and distance 40 mm using Exponent version 6.1.22.0, Stable Micro Systems, Surrey, UK. The curcumin dressings were supported using the film support rig, and a circular section of the film was exposed, allowing a spherical probe to be pushed through to perform extension and elasticity measurements (extensibility). During a test, the maximum force to rupture the sample (toughness) is recorded.

#### 3.2.3. Erosion Time

Curcumin-loaded wafer dressings were placed individually in 6 glass tubes of the basket assembly that was immersed in a thermo-stated water bath containing 1 liter (L) of phosphate buffer pH 7.4 containing 1% Tween 80 at 37 ± 1 °C using the conventional tablet disintegration tester Pharma Test PTZ-S, GmbH, Hainburg, Germany. The basket assembly (six-tubes) was raised and lowered at a frequency of 30 strokes/min. The endpoint for erosion time was recorded after complete erosion of the wafer dressings and passing through the mesh screen.

### 3.3. X-ray Diffraction (XRD)

The crystallinity of curcumin in the prepared wafer dressings was studied. X-ray diffraction patterns were obtained by XRD diffractometer (Unisantis XMD-300, GmbH, Germany). The tube operated at 45,000 V and 0.8 mA. The scan range was 5–60° with a diffraction angle of 2θ.

### 3.4. Differential Scanning Calorimetry (DSC)

Specific weights of curcumin, drug-free wafers, and drug-loaded wafers were placed in a 40 µL-capacity aluminum pan and hermetically sealed and pierced. The pan temperature was gradually raised from 30 °C to 350 °C at 10 °C/min using a DSC calorimeter (DSC-60; Shimadzu, Kyoto, Japan). Nitrogen was used as a purging gas at a flow rate of 20 mL/minute.

### 3.5. Fourier Transform Infrared Spectroscopic (FTIR) Study

FTIR spectrophotometry (FT-IR, Tensor 37, Bruker, Billerica, MA, USA) was used to stack the spectra of curcumin, drug-free wafers, and drug-loaded wafers. Spectra were collected from KBr discs. The spectra were collected in a range of 4000 to 400 cm^−^^1^, and number of scans per sample was set at 20.

### 3.6. In Vitro Release Studies

In vitro drug release was studied using a water-bath shaker (Shel Lab water bath, Sheldon, Cornelius, OR, USA) at 37 ± 0.5 °C at a shaking speed of 100 strokes per min. The donor compartment contained a wafer dressing (2 cm in diameter) or curcumin suspension and immersed in the receptor compartment. The receptor compartment was composed of a 250 mL beaker filled with phosphate buffer (pH 7.4) containing Tween 80 (0.1% *w*/*v*). A sample of 5 mL was withdrawn from the medium every hour, and an equal volume of fresh medium was added to keep the volume constant. A curcumin suspension (0.1%) was used as the control formulation. The absorbance was measured by Shimadzu UV-2550 Spectrophotometer, Kyoto, Japan, at 470 nm. This experiment was performed in triplicate. The cumulative release data were fitted to the most commonly used kinetics models (zero order, first order, Higuchi, and Korsmeyer–Peppas models) using KinetDS 2.0.

### 3.7. Wound Healing Study

#### 3.7.1. Animals

Twenty-five male Wistar albino rats (1–1.5 months old and weighing 140–190 g) were collected from the Central Animal House Facility, King Khalid University. The animals were caged in 5 groups (5 mice each) were acclimatized for two weeks, and fed with standard diet from local suppliers and free access to water (ad libitum). The experiment complied with the OECD guidelines, and the study was permitted by the Institutional Ethical Committee (Approval Ref. No: ECM 2022-2901) of King Khalid University, Saudi Arabia.

#### 3.7.2. Wound Incision Model and Treatment

The rats were randomly divided into 5 groups (the names of the groups are shown Table 5). The animals were anesthetized using 2% xylazine (5 mg/kg) and 10% ketamine (90 mg/kg) intraperitoneally. A circular wound of 2 cm in diameter was made on the dorsal thoracic region of rats and observed individually throughout the study [13,33]. Treatment was initiated on day zero by application of drug-free wafer dressings (the placebo group), curcumin suspension, F5, F6, and F7 every other day under occlusive conditions using Silk pool plaster (backing adhesive plaster) Sensitive Fix^®^, Pic solution, Artsana group, Italy. This dosage regimen was selected for being more convenient clinically by minimizing daily removal of the plaster and allowing minimal animal disruption and handling for better wound healing. The wound area was measured by tracing the wound on calibrated transparent sheets. Figure 9 shows representative photographs of dissected wound at day zero, application of the curcumin-loaded wafer dressing, and fixing of the applied wound dressing using the backing adhesive plaster.

### 3.8. Statistical Analysis

Statistical analysis was conducted using Graph Pad Prism software, from the USA, to test for analysis of variance (ANOVA) with a statistical significance set at *p* < 0.05.

## 4. Conclusions

Six out of seven curcumin-loaded wafer dressings were successfully prepared using the solvent casting method. The main selection criteria for further investigation and in vivo evaluation were the mechanical characteristics and faster erosion time for rapid wetting and drug release. Curcumin was dispersed in an amorphous form within the polymeric wafer dressings. Polymeric wafer dressings loaded with curcumin showed desirable strength and elasticity. The mechanical properties indicated that the prepared wafer could withstand handling by patients. Prolonged drug release over a period of >24 h was recorded with enhanced wound healing rates in Wistar albino rat models; superior wound healing was recorded at a dose size of 10 mg of curcumin per wafer dressing. These novel curcumin-loaded wafer dressings can be promising medicated wound dressings providing faster and safer wound healing rates.

## Figures and Tables

**Figure 1 pharmaceuticals-16-00038-f001:**
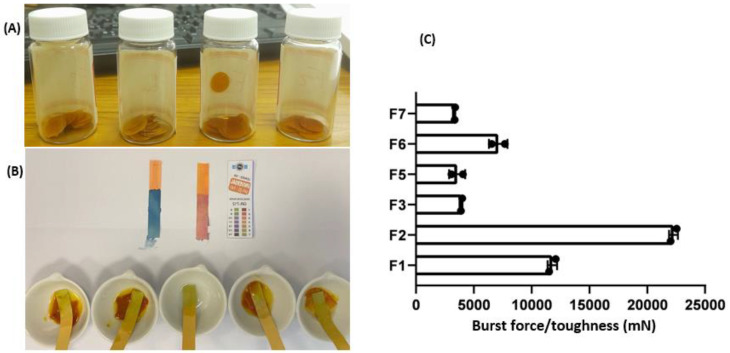
Curcumin loaded wafer dressings (**A**), the set-up of surface pH measurements (**B**) and measured toughness (burst force) for the prepared wafer dressings (**C**). Results represent mean ± SD, *n* = 6.

**Figure 2 pharmaceuticals-16-00038-f002:**
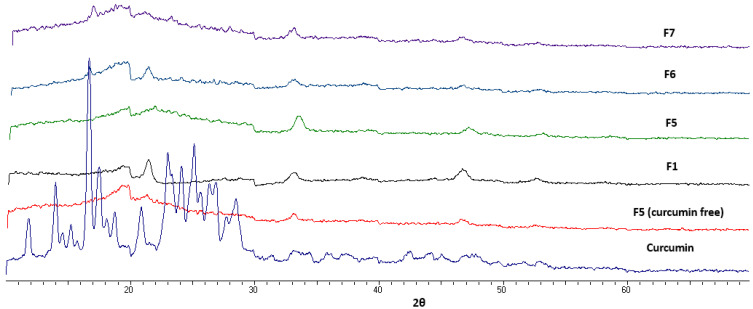
X-ray diffractograms of curcumin, drug-free wafer (F5) and some selected curcumin loaded wound dressing wafers (F1, F5–F7).

**Figure 3 pharmaceuticals-16-00038-f003:**
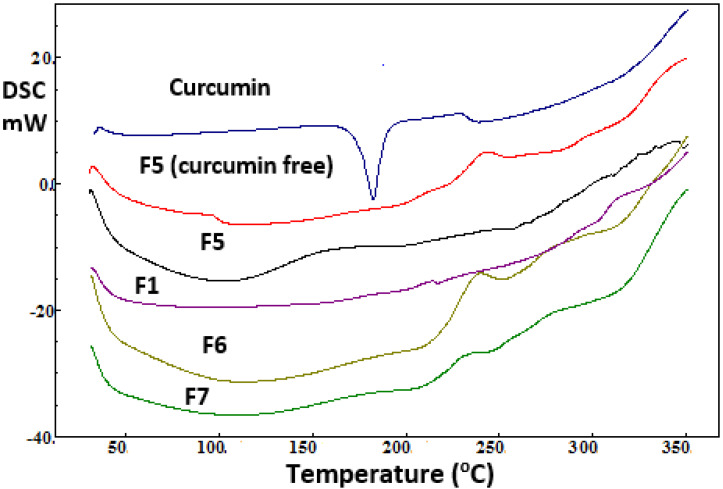
DSC thermograms of curcumin, drug-free wafer (F5) and some selected curcumin loaded wound dressing wafers (F1, F5–F7).

**Figure 4 pharmaceuticals-16-00038-f004:**
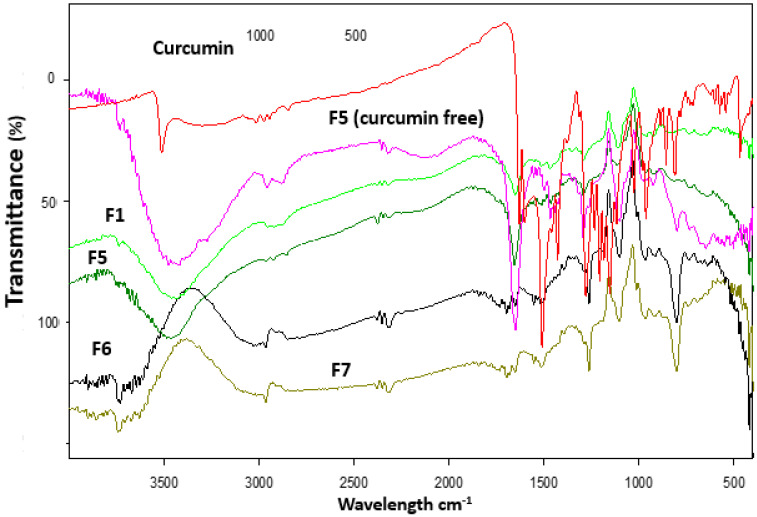
FTIR spectra of curcumin, drug-free wafer (F5) and some selected curcumin loaded wound dressing wafers (F1, F5–F7).

**Figure 5 pharmaceuticals-16-00038-f005:**
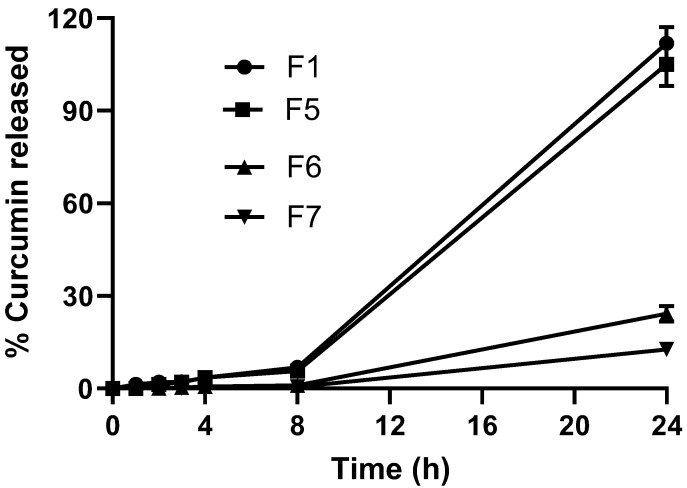
In vitro curcumin release from selected wafer dressings. Results represent mean ± SD, *n* = 3.

**Figure 6 pharmaceuticals-16-00038-f006:**
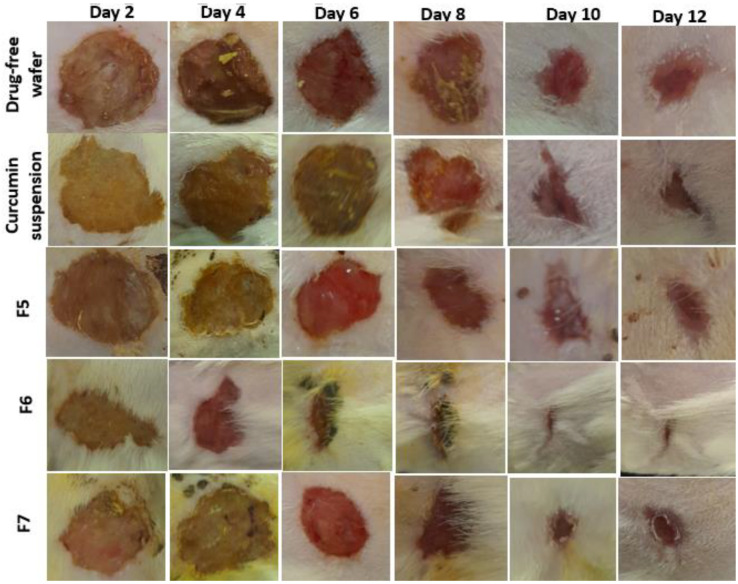
Photographs of wound healing processes in rats over a 12-days period for curcumin-free wafer dressing treated control, curcumin suspension-treated group and three different curcumin wound dressing formulations (F5–F7) containing three different concentrations of curcumin (x, 5-x and 10x-fold increments of curcumin).

**Figure 7 pharmaceuticals-16-00038-f007:**
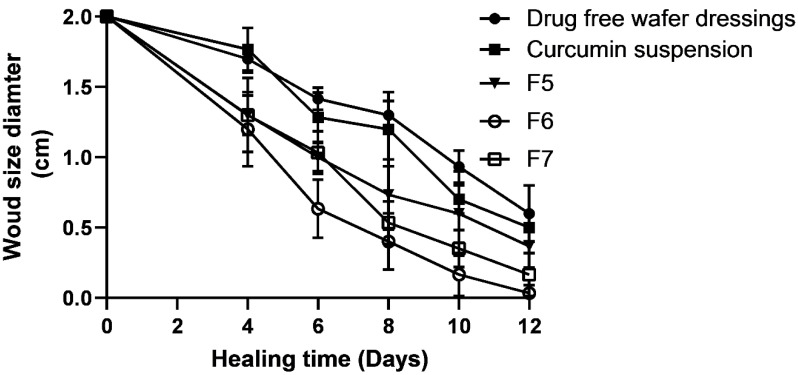
Wound healing rates (change in diameter in cm) over time for curcumin-free wafer dressing treated control, curcumin suspension-treated group and three different curcumin wound dressing formulations (F5–F7) containing three different concentrations of curcumin (x, 5x- and 10x-fold increments of curcumin). Data points represent mean ± SD, *n* = 5.

**Figure 8 pharmaceuticals-16-00038-f008:**
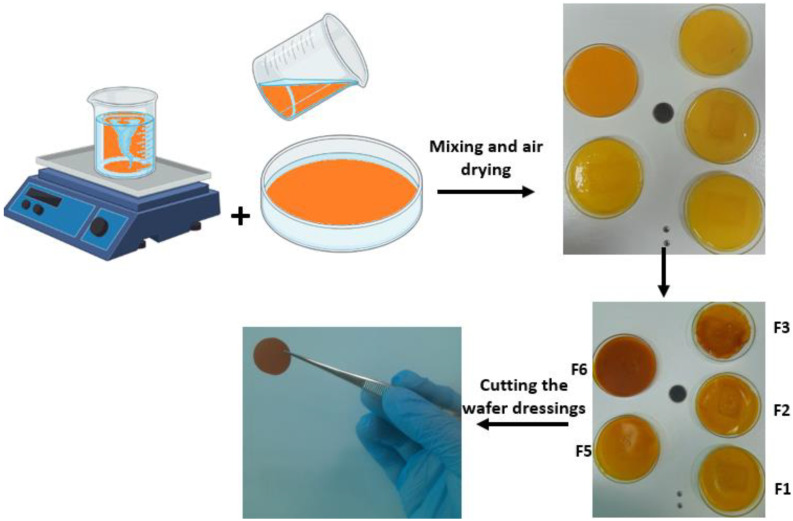
Graphical outline of the solvent casting method for preparation of curcumin-loaded wound dressings.

**Figure 9 pharmaceuticals-16-00038-f009:**
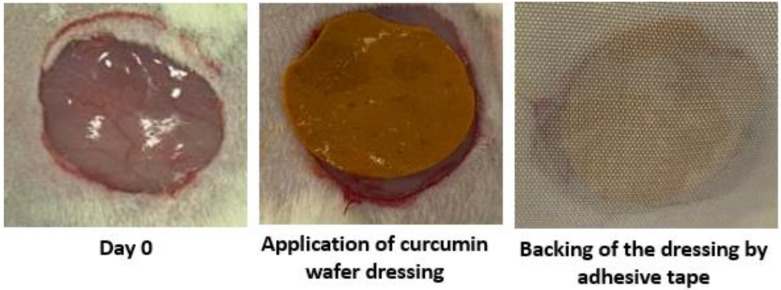
Representative photographs of the wound at day zero, application of the curcumin-loaded wafer dressing and backing of the wafer dressing using Sensitive Fix^®^ adhesive plaster.

**Table 1 pharmaceuticals-16-00038-t001:** Dimension, drug content, folding endurance and erosion time measured for the prepared curcumin-loaded wafer dressings. Results represent mean ± standard deviation (SD), *n* = 6.

Formulation	F1	F2	F3	F5	F6	F7
Content (mg)	2.0 ± 0.4	1.85 ± 0.3	1.9 ± 0.3	1.95 ± 0.3	10 ± 0.5	20 ± 1.0
Weight (mg)	112 ± 5.5	114 ± 6.5	115 ± 7.0	118 ± 8.0	130 ± 10.0	135 ± 12.0
Thickness (µm)	210 ± 10	200 ± 5.0	205 ± 15	210 ± 10	310 ± 5.5	325 ± 6.5
pH	7	7	7	7	7	7
Folding endurance	10 ± 3.0	10 ± 1.0	3 ± 1.0	10 ± 2.0	8 ± 2.0	6 ± 4.0
Toughness (mN)	12,078 ± 453	22,537 ± 541	3067 ± 321	39,067 ± 342	7112 ± 711	3894 ± 781
Extensibility (mm)	126 ± 13.0	125 ± 11.0	126 ± 1.0	125 ± 2.0	126 ± 2.0	123 ± 2.0
Erosion time (min)	30 ± 3.5	18 ± 4.0	5 ± 2.5	5 ± 1.5	10 ± 5	15 ± 2.0

**Table 2 pharmaceuticals-16-00038-t002:** Regression coefficient (R) and release rate constant (K) generated from the adopted kinetics models using KinetDS 2.0.

Formula	Zero		First		Higuchi		Korsmeyer–Peppas		
	R	K_0_	R	K	R	K_H_	R	K	*n*
F1	0.91	1.18	0.94	2.75	0.99	8.51	0.79	7.02	0.56
F5	0.8	1.20	0.91	1.42	0.94	5.24	0.81	3.8	0.55
F6	0.84	0.74	0.88	0.67	0.97	3.3	0.81	3.01	0.51
F7	0.77	0.71	0.81	0.53	0.94	2.45	0.91	1.89	0.52

**Table 3 pharmaceuticals-16-00038-t003:** Healing rate constants, half-life (T_1/2_) and regression coefficients for the wound closure estimated for curcumin-free wafer dressing(placebo)-treated control, curcumin suspension-treated group and three different curcumin wafer dressings (F5–F7) containing three different concentrations of curcumin (1×-, 5×- and 10×-fold increments of curcumin). Results represent mean ± SD, *n* = 5. * denotes statistical significant difference (*p* < 0.05); ** denotes no statistical significant difference (*p* > 0.05).

Formulation	Healing Rate (Day^−1^)	T_1/2_ (Days)	Regression Coefficient (R)
Drug-free wafer	0.032 ± 0.005	20 ± 3.0	0.95
Curcumin suspension	0.043 ± 0.003 *	13 ± 1.5 *	0.97
F5 (2 mg)	0.058 ± 0.002 *	10 ± 1.0 *	0.99
F6 (10 mg)	0.095 ± 0.007 *	7.0 ± 1.0 *	0.98
F7 (20 mg)	0.095 ± 0.007 **	8.5 ± 1.5 **	0.97

**Table 4 pharmaceuticals-16-00038-t004:** Composition of the prepared curcumin loaded wafer dressings.

Formulation	F1	F2	F3	F4	F5	F6	F7
Curcumin (mg)	34	34	34	34	34	170	340
HPMC 5% (g)	20	-	-	-	5	5	5
CMC 5% (g)	-	20	-	-	5	5	5
SA 5% (g)	-	-	20	-	5	5	5
PVP K15 10% (g)	-	-	-	10	2.5	2.5	2.5
PVA 5% (g)	5	5	5	5	5	5	5
Dextran (mg)	100	100	100	100	100	100	100
PEG 1000 (mg)	100	100	100	100	100	100	100
Glycerin (mg)	250	250	250	250	250	280	310

**Table 5 pharmaceuticals-16-00038-t005:** Animal groups, treatment and dosage received.

Groups (*n* = 5)	Treatment/Dosage
Group I	Placebo/drug free-wafer dressing applied every other day
Group II	Curcumin suspension (10 mg/mL)/1 mL applied every other day
Group III	F5 applied every other day
Group IV	F6 applied every other day
Group V	F7 applied every other day

## Data Availability

Data is contained within the article.

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
