# Peer review of "Polymeric/Dextran Wafer Dressings as Promising Long-Acting Delivery Systems for Curcumin Topical Delivery and Enhancing Wound Healing in Male Wistar Albino Rats"

_pharmaceuticals, 2022, doi:10.3390/ph16010038_

Round 1

Reviewer 1 Report

The manuscript entitled “Polymeric/dextran wafer dressings as promising long-acting delivery systems for curcumin topical delivery and enhancing wound healing in male Wistar albino rats” showed interesting data for development of curcumin-loaded wafers in wound-healing applications. However, there are some comments and questions about this manuscript as follows:

1. The authors should specify the novelties of this research in the introduction section.

2. The authors should inform the difference between of polymeric wafer dressings and polymeric patches for wound healing in the introduction section.  In addition, the advantages of polymeric wafer dressing over patches should be mentioned in this section.

3. In Table 1, the authors should add some details for each parameters. For example: contents of curcumin/wafer, net weight/wafer, thickness/wafer.

4. To determine the erosion time, the authors should specify the end point of complete erosion of the wafers from using the disintegration tester in the materials and methods section.

In addition, were the disintegration discs put into the disintegration basket after the wafers were loaded? Please inform the details in the materials and methods section.

5. Due to the poor solubility of curcumin, the slopes of the release profiles of curcumin from the wafers during 0-8 hours were very low.  This finding is usually found when hydrophobic drugs have been subjected to the release study.

However, the authors should continuously collect the receiving medium with more frequencies since the 8th to 24th, for example, every 1-2 hours, until the experiment was finished.  This procedure could maintain the sink conditions during the release study was performed.

6. The authors should inform the kinetics of drug release from the wafers and inform the benefits of their drug release kinetic in delivery of curcumin for wound healing.

7. In the section 3.6., in vitro release studies, the authors stated that a curcumin suspension was also included in this study, why was its release profile not shown in Fig. 5?

Since the authors use the suspension of curcumin as one of test samples in the in vivo wound healing experiment, the release profile of the curcumin suspension should be shown in the manuscript.

8. The authors should inform in the section 3.6, in vitro release profile, what the donor compartment and the receptor compartment were.

9. Why were the test samples applied on the wounds every other day (not every day)? Please inform the reason in the materials and methods section.

10. In the section 3.7 Wound healing study, the authors stated that the animals were caged in 6 groups.  However, from the results, they were actually divided into 5 groups according to the number of test samples.  Could the authors verify this information? If they were caged in 6 groups, what did the authors do with the rest group? Please inform in the section 3.7.

Author Response

We appreciate your constructive comments which are all valuable and very helpful for revising and improving our paper, as well as guiding our research. We have carefully considered comments and have revised the manuscript accordingly. Our point-by-point responses to the comments are listed below.

The manuscript entitled “Polymeric/dextran wafer dressings as promising long-acting delivery systems for curcumin topical delivery and enhancing wound healing in male Wistar albino rats” showed interesting data for development of curcumin-loaded wafers in wound-healing applications. However, there are some comments and questions about this manuscript as follows:

  1. The authors should specify the novelties of this research in the introduction section.

-Novelty aspects have been highlighted in the introduction section.

  1. The authors should inform the difference between of polymeric wafer dressings and polymeric patches for wound healing in the introduction section. In addition, the advantages of polymeric wafer dressing over patches should be mentioned in this section.

-Differences between patches and wafer have been described in the introduction section Line (47-51) and the advantages in the introduction section line (52-68).

  1. In Table 1, the authors should add some details for each parameters. For example: contents of curcumin/wafer, net weight/wafer, thickness/wafer.

-The section has been modified accordingly.

  1. To determine the erosion time, the authors should specify the end point of complete erosion of the wafers from using the disintegration tester in the materials and methods section.

In addition, were the disintegration discs put into the disintegration basket after the wafers were loaded? Please inform the details in the materials and methods section.

-The section has been modified accordingly.

  1. Due to the poor solubility of curcumin, the slopes of the release profiles of curcumin from the wafers during 0-8 hours were very low. This finding is usually found when hydrophobic drugs have been subjected to the release study.

However, the authors should continuously collect the receiving medium with more frequencies since the 8th to 24th, for example, every 1-2 hours, until the experiment was finished.  This procedure could maintain the sink conditions during the release study was performed.

-The sink condition was maintained throughout the in vitro release experiment, the release medium contained surfactant and pH 7.4 allows ionization of curcumin. The results and discussion section has now been modified.

  1. The authors should inform the kinetics of drug release from the wafers and inform the benefits of their drug release kinetic in delivery of curcumin for wound healing.

-Table.2 has now been inserted to show the kinetics models and parameters to explain the release mechanism.

  1. In the section 3.6., in vitro release studies, the authors stated that a curcumin suspension was also included in this study, why was its release profile not shown in Fig. 5?

Since the authors use the suspension of curcumin as one of test samples in the in vivo wound healing experiment, the release profile of the curcumin suspension should be shown in the manuscript.

-The release of curcumin suspension was too slow to be shown on the curve. The section has now been modified and stated in the results section.

  1. The authors should inform in the section 3.6, in vitro release profile, what the donor compartment and the receptor compartment were.

-This section has been modified to clearly show the donor and receptor compartments.

  1. Why were the test samples applied on the wounds every other day (not every day)? Please inform the reason in the materials and methods section.

-Section 3.7 has now been modified accordingly.

  1. In the section 3.7 Wound healing study, the authors stated that the animals were caged in 6 groups. However, from the results, they were actually divided into 5 groups according to the number of test samples. Could the authors verify this information? If they were caged in 6 groups, what did the authors do with the rest group? Please inform in the section 3.7.

-This is a valid point and the authors thank the reviewers to that. The number of groups has been corrected to five.

Reviewer 2 Report

Overall, the work is promising, but some questions need to be answered.

1. In the Abstract, I recommend highlighting the main objective of the work.

2. In the "Materials" section, it lists the polymers, but for sodium alginate, it did not provide any manufacturer information, as in the other cases. Also enter molecular weight or viscosity data here.

3. During the FTIR measurements, how many scans were performed per measurement and what was used as background?

4. Why did you use Tween 80 in the receptor compartment for the in vitro release studies?

5. The first section of the „Results” section should be reconsidered, whether it is in the right place, or rather the most important results should be inserted in the Conclusion section from here.

6. How did you measure elasticity?

7. How did you calculate the tensile strength? (131 lines) If you have actually calculated it, you should include the results in the article.

8. The quality of Figure 3 should be improved. The numbers on the axes should be clearly visible. The figure shows 6 curves, but there are only 5 legends in the figure, one is missing. I would recommend deleting the title "Curcumin", it is not necessary for the figure. Instead of "F5 (plain)", the name "F5 (curcumin free)" should be used, as in the XRD diagram, in order to be consistent. The y-axis should be normalized to mass.

9. The quality of Figure 4 should also be improved. Please check the y-axis assignment, numbering and direction. Why is the y-axis reversed? If this is really transmittance, should the axis be between 0-100%? I would recommend deleting the title "Curcumin", it is not necessary for the figure.

10. In the case of the Skin test, why did you not investigate the F1 sample? Based on previous investigations, he wrote that it did not rule this out from further investigations.

11. Table 2 is not referred to in the text. Please replace it.

12. How did you calculate the "healing rate"?

13. On what basis did you compare the mucoadhesivity of different polymer films? Have you performed mucoadhesion strength measurements? In the article below, it was reported that HPMC has greater mucoadhesive strength than sodium alginate:  Pharmaceutics 2021, 13(5), 619; https://doi.org/10.3390/pharmaceutics13050619 In your study, you found exactly the opposite. What could be the reason for this?

Author Response

We appreciate your constructive comments which are all valuable and very helpful for revising and improving our paper, as well as guiding our research. We have carefully considered comments and have revised the manuscript accordingly. Our point-by-point responses to the comments are listed below.

Overall, the work is promising, but some questions need to be answered.

Thank you for that.

  1. In the Abstract, I recommend highlighting the main objective of the work.

The main objective has been highlighted in the abstract

  1. In the "Materials" section, it lists the polymers, but for sodium alginate, it did not provide any manufacturer information, as in the other cases. Also enter molecular weight or viscosity data here.

The manufacture information and viscosity data have been provided for sodium alginate

  1. During the FTIR measurements, how many scans were performed per measurement and what was used as background?

Background noise was eliminated and number of scans was 20. The method section has now been modified accordingly.

  1. Why did you use Tween 80 in the receptor compartment for the in vitro release studies?

The purpose of tween 80 was mentioned in lines 211-212 to ensure proper wetting of curcumin wafer dressings and maintain sink conditions.

  1. The first section of the „Results” section should be reconsidered, whether it is in the right place, or rather the most important results should be inserted in the Conclusion section from here.

 The first few lines under section 2.1. are thought to be important to outline the design and rationale of using these polymers.

  1. How did you measure elasticity?

The elasticity is measured using two experiments one empirical (folding endurance) and the other one is mechanical Texture analyzer by determining extensibility.

  1. How did you calculate the tensile strength? (131 lines) If you have actually calculated it, you should include the results in the article.

The tensile strength was measured using the texture analyzer by measuring the maximum force to rupture the sample (toughness). The results are outlined in Table 1.

  1. The quality of Figure 3 should be improved. The numbers on the axes should be clearly visible. The figure shows 6 curves, but there are only 5 legends in the figure, one is missing. I would recommend deleting the title "Curcumin", it is not necessary for the figure. Instead of "F5 (plain)", the name "F5 (curcumin free)" should be used, as in the XRD diagram, in order to be consistent. The y-axis should be normalized to mass.

This is a valid point and the authors thank the reviewer for this. The resolution of the figure has been enhanced to 300 DPI. Curcumin was shifted to be clearly indicate a legend for curcumin powder So there are 6 legends. The F5 (plain) was changed to F5 (curcumin free).

  1. The quality of Figure 4 should also be improved. Please check the y-axis assignment, numbering and direction. Why is the y-axis reversed? If this is really transmittance, should the axis be between 0-100%? I would recommend deleting the title "Curcumin", it is not necessary for the figure.

The quality of Figure 4 has now been enahcned and the Y-axis and curcumin legend has now been adjusted.

  1. In the case of the Skin test, why did you not investigate the F1 sample? Based on previous investigations, he wrote that it did not rule this out from further investigations.

F5 showed superior loading capacity to accommodate larger doses of curcumin. For comparison reasons, F5, F6 and F7 had the same composition but different drug doses.

  1. Table 2 is not referred to in the text. Please replace it.
  2. Table 2 becomes Table 3 and it has now been cited in the text (line 234)

  1. How did you calculate the "healing rate"?

The heling rate was expressed as healing rate constant and time for 50% reduction in wound size. These parameters were estimated through fitting into first order equation.

  1. On what basis did you compare the mucoadhesivity of different polymer films? Have you performed mucoadhesion strength measurements? In the article below, it was reported that HPMC has greater mucoadhesive strength than sodium alginate:  Pharmaceutics 2021, 13(5), 619; https://doi.org/10.3390/pharmaceutics13050619 In your study, you found exactly the opposite. What could be the reason for this?

We have published previous work on ocular films containing HPMC and other polymers and mucoadhesion was tested. HPMC is always less mucoadhesive compared with CMC and SA. This is because CMC and SA contained carboxyl groups that engage in stronger electrostatic attraction and  H-bonding with mucus. These two references were provided.

Reviewer 3 Report

The authors are appreciated to attempt a novel approach and a new way of drug delivery systems ...But

The manuscript scientific soundness will be improved

Scientific justification may be modified

Result and discussion parts should be rewritten with more references or citations 

Such as the selection and need of the studies also the stability of the selected drug substance ...Likely  

Author Response

We appreciate your constructive comments which are all valuable and very helpful for revising and improving our paper, as well as guiding our research. We have carefully considered comments and have revised the manuscript accordingly. Our point-by-point responses to the comments are listed below.

The authors are appreciated to attempt a novel approach and a new way of drug delivery systems ...But

The manuscript scientific soundness will be improved

Scientific justification may be modified.

Introduction and parts of results and discussion have been modified accordingly.

Result and discussion parts should be rewritten with more references or citations 

Such as the selection and need of the studies also the stability of the selected drug substance ...Likely  

References have now been added whenever possible. The stability of the prepared formulation has been investigated because it was beyond the scope of this work.

Reviewer 4 Report

Some comments that the authors need to be addressed 

1) it would be better if table one could be demonstrated as a bar curve, which could more clearly demonstrate the difference between the different parameters.

2) The FTIR curve in figure 4 should be separated as the independent curve without overlapping each other.

3) I could not find the error bar in figure 5; please recheck it.

4) The scale bar is lost in figure 5, and no statistical difference in wound size diameter.

Author Response

We appreciate your constructive comments which are all valuable and very helpful for revising and improving our paper, as well as guiding our research. We have carefully considered comments and have revised the manuscript accordingly. Our point-by-point responses to the comments are listed below.

Some comments that the authors need to be addressed

1) it would be better if table one could be demonstrated as a bar curve, which could more clearly demonstrate the difference between the different parameters.

-The manuscript contains 9 figures which suffices the required number of figures and there were no statistical differences among recorded values for weight, content and thicknesses.

2) The FTIR curve in figure 4 should be separated as the independent curve without overlapping each other.

For comparative purposes, it would be better to stack the FTIR spectra. The overlapping is only in the finger print region. The individual curves have now been submitted as supplementary mate

3) I could not find the error bar in figure 5; please recheck it.

    -Figure 5 has now been modified accordingly.

4) The scale bar is lost in figure 5, and no statistical difference in wound size diameter.

-The statistical differences of wound sizes among formulations and placebo-treated groups and between the formulations have been indicated in Table 3.

Round 2

Reviewer 1 Report

The authors corrected the manuscript following the suggestions from the reviewers.

Therefore, this manuscript can be accepted for publishing in this journal.

Author Response

The authors would like to thank the reviewer for the positive comments.

Reviewer 2 Report

Based on the responses and review of the revised manuscript, there are still questions and comments that need to be clarified before publication.

The term tensile strength is mentioned 2 times in the text, although the results are not found. I assume it could be a typo. Please correct it, this is not a synonymous term. Alternatively, the measured forces must be divided by the area and indicated in the manuscript. In addition, the nomenclature of the measured quantities should also be reconsidered (in your text: “burst force, toughness, extensibility, elongation”). For example, in Table 1, the term extensibility is used, while in Figure 1, force is listed, although it is probably the same quantity. In the literature, for example, the same texture analyzer was used in the following article: https://doi.org/10.1016/j.ijpharm.2013.11.033, but the nomenclature of the measured quantities is different. Furthermore, in the Methods section, it is necessary to define exactly what is meant by each measured quantity. The term elongation is also mentioned twice in the text, but at the same time it is not defined, nor can the results be found. This is not a synonymous term. It could also be a typo, please standardize the nomenclature.

Figure 3 is still inappropriate for a high-level scientific journal. Please correct the numbers on the axes. Either delete the zeros after the decimal point or make both zero characters visible in their entirety. I recommend the former. Furthermore, the inscription "Temperature" on the x-axis should be corrected instead of "Temperture".

The terms "First" and "Zero" in the header of Table 2 should be replaced by "First order" and "Zero order" for clarity. In the same table, I recommend using "Coefficient of determination" (R2) instead of "Regression coefficient" or "Coefficient of correlation" (R) terms. Similarly in Table 3.

The calculation method of "Healing rate" must also be clearly stated in the manuscript and what exactly it means must be defined.

Author Response

The term tensile strength is mentioned 2 times in the text, although the results are not found. I assume it could be a typo. Please correct it, this is not a synonymous term. Alternatively, the measured forces must be divided by the area and indicated in the manuscript.

This is a valid point. The tensile strength was replaced with real parameter measured (extensibility).

In addition, the nomenclature of the measured quantities should also be reconsidered (in your text: “burst force, toughness, extensibility, elongation”). For example, in Table 1, the term extensibility is used, while in Figure 1, force is listed, although it is probably the same quantity. In the literature, for example, the same texture analyzer was used in the following article: https://doi.org/10.1016/j.ijpharm.2013.11.033, but the nomenclature of the measured quantities is different.

Figure 2 has now been modified accordingly force was replaced with burst force/toughness in the figure 2-C. Extensibility and toughness are two parameters named by the manufacturer catalogue for the spherical probe used.

Furthermore, in the Methods section, it is necessary to define exactly what is meant by each measured quantity. The term elongation is also mentioned twice in the text, but at the same time it is not defined, nor can the results be found. This is not a synonymous term. It could also be a typo, please standardize the nomenclature.

The word elongation has now been taken off.

Figure 3 is still inappropriate for a high-level scientific journal. Please correct the numbers on the axes. Either delete the zeros after the decimal point or make both zero characters visible in their entirety. I recommend the former. Furthermore, the inscription "Temperature" on the x-axis should be corrected instead of "Temperture".

Figure 3 has now been corrected accordingly.

The terms "First" and "Zero" in the header of Table 2 should be replaced by "First order" and "Zero order" for clarity. In the same table, I recommend using "Coefficient of determination" (R2) instead of "Regression coefficient" or "Coefficient of correlation" (R) terms. Similarly in Table 3.

Table 2 and Table 3 have been modified accordingly.

The calculation method of "Healing rate" must also be clearly stated in the manuscript and what exactly it means must be defined.

The method of estimating healing rate constants was clearly mentioned under Method Section 3.7 line 385-388.

Reviewer 3 Report

The manuscript may be improved with still proper scientific justifications 

Author Response

The authors thank the reviewer for the valuable comments and feedback. The manuscript has now been revised and method sections, figures and parts of discussion have been edited accordingly.

Reviewer 4 Report

all of my comments has been addressed, this paper can be accept for publishing.

Author Response

The authors pleased to hear we have addressed all comments . 

Round 3

Reviewer 3 Report

The result part must be improved with scientific justifications

Also the authors recommended to some similar papers to rewrite the result and discussion   

to refer 

Author Response

The authors would like to experess sincere appreciation to the time and rigorous reviewing that will contribute to enhacing the output of the manuscript. Responses to the comments were highlighted as track-change in the manuscript. Point-to-pont reposnes as below: 

The result part must be improved with scientific justifications

Every section in the results and discussion section has been revised and enhanced for scieitifc justifcation.

Also the authors recommended to some similar papers to rewrite the result and discussion   

Additional citations and references have been added to the manuscript in the results and discussion section.